# Risk factors for increased COVID-19 case-fatality in the United States: A county-level analysis during the first wave

Jess A. Millar[1]*, Hanh Dung N. Dao[2], Marianne E. Stefopulos[3], Camila G. Estevam[4], Katharine Fagan-Garcia[5], Diana H. Taft[6], Christopher Park[7], Amaal Alruwaily[8], Angel N. Desai[9]*, Maimuna S. Majumder[10]*

**1** Department of Epidemiology, Department of Computational Medicine and Bioinformatics, University of Michigan, Ann Arbor, MI, United States of America, **2** Department of Biostatistics and Epidemiology, University of Oklahoma Health Sciences Center, Oklahoma City, OK, United States of America, **3** Child Health Evaluative Sciences Program, SickKids Hospital, Toronto, Canada, **4** Department of Public Health, State University of Campinas, Campinas, SP, Brazil, **5** Department of Medicine, University of Alberta, Edmonton, AB, Canada, **6** Department of Food Science and Technology, University of California Davis, Davis, CA, United States of America, **7** College of Global Public Health, New York University, New York, NY, United States of America, **8** Independent Scholar, Riyadh, Saudi Arabia, **9** Division of Infectious Disease, Department of Internal Medicine, University of California Davis Medical Center, Sacramento, CA, United States of America, **10** Harvard Medical School and Boston Children's Hospital, Boston, MA, United States of America

\* jamillar@umich.edu (JAM); andesai@ad3.ucdavis.edu (AND); maimuna.majumder@childrens.harvard.edu (MSM)

## Abstract

The ongoing COVID-19 pandemic is causing significant morbidity and mortality across the US. In this ecological study, we identified county-level variables associated with the COVID-19 case-fatality rate (CFR) using publicly available datasets and a negative binomial generalized linear model. Variables associated with decreased CFR included a greater number of hospitals per 10,000 people, banning religious gatherings, a higher percentage of people living in mobile homes, and a higher percentage of uninsured people. Variables associated with increased CFR included a higher percentage of the population over age 65, a higher percentage of Black or African Americans, a higher asthma prevalence, and a greater number of hospitals in a county. By identifying factors that are associated with COVID-19 CFR in US counties, we hope to help officials target public health interventions and healthcare resources to locations that are at increased risk of COVID-19 fatalities.

## Introduction

The Severe Acute Respiratory Syndrome Coronavirus 2 (SARS-CoV-2) originated in Wuhan, China in November 2019 and has since spread to 210 countries worldwide [1]. By the last day considered for inclusion in the present work, June 12th, 2020, SARS-CoV-2 had caused over 2 million Coronavirus Disease 2019 (COVID-19) cases and 114,753 deaths in the United States (US) [2, 3]. The excess mortality from COVID-19 is likely to be underestimated, and recent

**Data Availability Statement:** The data underlying the results presented in the study are available

from (cited within the manuscript) https://github.com/jmillar201/COVID19_CFR.

**Funding:** This study was supported by the National Science Foundation in the form of a grant awarded to JAM (DGE-1256260) and the National Institute of Child Health and Human Development in the form of grants awarded to DHT (1F32HD093185-03) and MSM (T32HD040128). The funders had no role in study design, data collection and analysis, decision to publish, or preparation of the manuscript.

**Competing interests:** The authors have declared that no competing interests exist.

work estimates 912,345 deaths from COVID-19 between March 2020 and May 2021, compared to the officially reported 578,555 deaths [4]. Early work on COVID-19 has highlighted patient characteristics that increase an individual's risk of death [5, 6], however it is unclear to which individual risk factors are best suited to understanding which populations are most at risk of high fatality rates from COVID-19. The distribution of infected cases and fatalities in the US has been heterogeneous across counties [7], and identification of sub-populations at risk of increased morbidity and mortality remains crucial to effective response efforts by federal, state, and local governments [8]. Counties where governing officials are aware that their populations are at a higher risk of COVID-19 mortality, meaning the population experiences a higher case-fatality rate, may opt to tailor state policies or take earlier action to curtail the spread of SARS-CoV-2. Additionally, the federal government may opt to target vaccine resources to counties experiencing higher COVID-19 mortality rates.

The case-fatality rate (CFR) is defined as the number of deaths divided by the total number of confirmed cases from a given disease [9]. When a disease is non-endemic, the CFR fluctuates over time. During the beginning of an epidemic, there is often a lag when counting the number of deaths compared to cases and hospitalizations, leading to an underestimation of the CFR. Furthermore, CFR will fluctuate rapidly early in an epidemic when each additional case or death has an excessive impact on calculating CFR. It is important to not only account for the lag between cases and deaths (i.e., lag-adjusted CFR), but also to ensure that the CFR is no longer fluctuating.

In this study, our objective was to use a lag-adjusted CFR to conduct a county-level mortality risk factor analysis of demographic, socioeconomic, and health-related variables in the US during the first wave of the COVID-19 pandemic (March 28, 2020 to June 12, 2020). This will provide critical information on what population characteristics are most informative to identify counties at high risk of experiencing high COVID-19 mortality rates. We expand upon prior work by considering possible risk factors of an increased CFR from multiple categories (e.g., non-pharmaceutical interventions such as shelter-in-place orders [10]. prevalence of pre-existing conditions such as cardiovascular disease [11], and socio-economic circumstances such as hospital accessibility [12]) in a single model. This is also the first paper to focus on this range of risk factors during the first wave of the pandemic, so that results from this can be used for targeted intervention at the county level at the beginning of a pandemic.

## Methods

All code for our work can be found on our GitHub repository [13].

### Study population

We conducted a cross-sectional ecological study to assess risk factors associated with an increased COVID-19 lag-adjusted CFR in US counties. Our study population included 3,004 counties or county-equivalents with Federal Information Processing Standards (FIPS), a unique code for US federal identification (Appendix A1 in S1 Appendix). Only publicly available aggregate data were used; therefore, no IRB approval was required.

### County-level variables

We identified potential risk factors across several different categories: demographic, socioeconomic, healthcare accessibility, comorbidity prevalence, and non-pharmaceutical interventions. Each category-targeted risk factor relevant to the risk of COVID-19 mortality by conducting a comprehensive review of existing literature by March 28, 2020 supplemented with variables relevant to other respiratory epidemics [14–16]. Appendix A2 in S1 Appendix

provides detailed justifications for the inclusion of each risk factor. Only variables with publicly available data sources at the county- or state-level were included. Appendix A3 in S1 Appendix listed data sources, variable descriptions, and manipulations (if applicable). We directly imported and cleaned the datasets using R (v3.6.3).

We included five demographic variables: total population, population density, the percentage of the population over age 65, the percentage of population 17 or younger, and race/ethnicity. All demographic variable data were from the 2018 American Community Survey 5-Year Data from the US Census annual survey, except for race/ethnicity data from the U.S. Census Populations with Bridged Race Categories [17].

We included 13 socioeconomic variables, with their data primarily from the 2018 American Community Survey 5-Year Data [18]. In addition to the commonly used socioeconomic variables, we included certain variables contributing to the composite Social Vulnerability Index (SVI). The SVI was created by the Centers for Disease Control and Prevention (CDC) to describe US geographic areas by their social vulnerability and has been validated by multiple studies within and outside of the CDC [19–24] Social vulnerability is defined as "the characteristics of a person or community that affect their capacity to anticipate, confront, repair, and recover from the effects of a disaster." [19] We included individual SVI variables on socioeconomic status, household composition and disability, minority status and language, and housing and transportation. We preferred to use the individual variables rather than overall SVI or by theme because we were most interested in understanding which components of social vulnerability contributed to increased CFR.

We included 5 healthcare-related variables: number of hospitals per capita, number of ICU beds per capita, number of primary care physicians per capita, percentage of residents without health insurance, and percentage of Medicaid eligible residents. Variable data was from the Kaiser Health News [25], the Heart Disease and Stroke Atlas [26], and the 2018 American Community Survey 5-Year Data [18].

We included 18 comorbidity variables: diagnosed diabetes prevalence; diagnosed obesity prevalence; hypertension hospitalization and death prevalence, cardiovascular disease (CVD), chronic obstructive pulmonary disease (COPD), asthma, and cancer; Medicare beneficiaries with heart disease percentage, current smokers prevalence, and stroke-related hospitalization and mortality prevalence. Variable data was from the US Diabetes Surveillance System [27], the Heart Disease and Stroke Atlas [26], the Behavioral Risk Factor Surveillance System [28], and the State Cancer Profiles by the National Cancer Institute [29].

Non-pharmaceutical intervention data (including information on closing of public venues such as restaurants, gathering size limits, complete lockdown of non-essential activity in the county, if religious gatherings were included in gathering size limits, shelter-in-place orders, and social distancing mandates) were extracted from the COVID-19-intervention GitHub page, an open source data-sharing platform and compiled by Keystone Strategy [30]. However, this resource does not cover all counties, thus missing data was supplemented from a variety of governmental executive orders and news articles detailed in the supplementary code. Variables with dates were transformed to how many days the event occurred after the first case in a county. States where an intervention never occurred were given a zero. Since 47% of all counties did not ban religious gatherings, data on when religious gatherings were banned in a county was transformed into an indicator variable (1 if the ban occurred, 0 if not).

## Lag adjusted case-fatality rate (CFR) data and calculation

To calculate CFR during the first COVID-19 wave in the US, we obtained open access county-level COVID-19 data from the New York Times through June 12, 2020, the date the CDC

released guidance for easing restrictions as states began to reopen [2, 31]. Only data that contained FIPS county codes to identify case and death locations were included. County-level data for New York City, NY was accessed from the New York City Department of Health and Mental Hygiene [32]. To calculate lag-adjusted CFR (laCFR), we used Nishiura et al.'s method, expanded upon by Russell et al., to account for the delay between COVID-19 diagnoses and deaths [33, 34]. We updated this approach by using time-from-hospitalization-to-death from the US population [34, 35]. The final dataset included 1,779 counties with 1,968,739 cases and 106,279 deaths, comprising 96.8% of national cases and 96.8% of national deaths as of June 12, 2020.

During the first wave of the pandemic, SARS-CoV-2 was non-endemic, leading the case-fatality rate (CFR) to fluctuate over time. This is due to a lag when counting the number of deaths compared to cases and hospitalizations, leading to an underestimation of the CFR. The CFR continues to fluctuate rapidly early in an epidemic when each additional case or death has an excessive impact on calculating CFR. It is important to not only account for the lag between cases and deaths (i.e., lag-adjusted CFR), but also to ensure that the CFR is no longer fluctuating.

To do this, we use a method developed by Nishiura et al. and expanded upon by Russell et al., where case and death incidence data are used to estimate the number of cases with known outcomes, i.e. cases where the resolution, death or recovery, is known to have occurred [33, 34]:

$$u_t = \frac{\sum_{i=0}^{t} \sum_{j=0}^{\infty} c_{i-j} f_i}{\sum_{i=0}^{t} c_j}$$

where $c_t$ is the daily case incidence at time $t$, (with time measured in calendar days), $f_t$ is the proportion of cases with delay $t$ between onset or hospitalization and death; $u_t$ represents the underestimation of the known outcomes and is used to scale the value of the cumulative number of cases in the denominator in the calculation of the laCFR. Russell et al. used the estimated distribution in Linton et al., based on data from China up until the end of January 2020. For this study, we instead used United States centric data from Lewnard et al., which estimates the distribution of time from hospitalization to death based on data from Washington and California [35].

Lewnard et al., fits the distribution conditionally on age resulting in a Weibull distribution for each age group [35]. The overall distribution was obtained empirically by weighting the densities at time t across all age groups. Because of this, the overall distribution doesn't have its own shape/scale parameters. However, we were able to estimate what these parameters would be by fitting a Weibull distribution that captures the 2.5, 25, 50, 75, and 97.5 percentiles (1.6, 7.3, 12.7, 19.8, 37.4), as well as the average (14.5).

Use of the laCFR assumes the measure has stabilized [33]. Counties where the laCFR is still rapidly changing cannot be used in the study as these are not unbiased estimates of the true CFR. laCFRs were calculated incrementally for each day and assessed whether they changed on average less than 1% a week for the last two weeks of available data. The final calculation based on all data available was used as the laCFR in our model.

## Statistical analysis

To reduce multicollinearity, we eliminated linear combinations and variables with correlations >0.5 using the R package caret (v6.0.86). Remaining variables were screened for missingness and missing values were imputed using five imputations in the R package mice (v3.8.0) [36].

Data were randomly split into training (1,186 counties) and testing sets (593 counties) to assess generalizability (a table of the characteristics can be seen in Appendix A4 in S1 Appendix). A negative binomial linear model with an offset for the number of COVID19 cases per county was chosen based on Kolmogorov-Smirnov and dispersion tests found in the R package DHARMa (v0.3.1). Variable selection was conducted using purposeful selection, an iterative process in which covariates are removed from the model if they are neither significant nor confounders [37, 38]. With clinical risk factors, purposeful selection outperforms other variable selection procedures and tests for the presence of confounders [37]. Removing highly correlated variables beforehand reduces the chance of multicollinearity between non-significant variables that may have been retained in purposeful selection due to confounding effects. Per Bursac et al., we used the 0.1 α-level for initial selection using bivariate models and a change of >20% in any remaining model coefficients compared with the full multivariate model for confounding evaluation [37]. All variables in the final model were significant at the 0.05 α-level, and no statistical confounders were included in the final model.

## Model fit

We observed the fit of the model using a half-normal plot (Fig 1A). The simulated envelope for the deviance residuals in the half-normal plot serves as a guide of what to expect under a well-fitted model, with most of our model's deviance residuals lying within [39]. We compared the mean and variance seen within our model predictions to the theoretical mean and variance expected in a Poisson and negative binomial model. After grouping the fitted predictions into 20 quantiles and calculating their means and variances, we saw the negative binomial model captures our data variance well [40]. Loess smooth was used for the empirical mean (Fig 1A). As an additional check, we calculated the ratio of Pearson residuals to degrees of freedom, which was 1.04, indicating we accounted for most of the over-dispersion in laCFR using the negative binomial model. The Cox and Snell Pseudo R2 for our model was 0.86, which accounts for the majority of the variance present in our outcome variable. All variables had a variance inflation factor of less than 2, indicating collinearity was not an issue with our variables (Appendix 3 in S1 Appendix).

We checked the coverage, which is the probability that our model outcomes are found within our prediction interval. To estimate our predictive coverage (empirical coverage), we simulated a prediction interval. The coverage was 0.9730 for the training data and 0.9713 for testing data (Fig 2A). Similar to ROC, a gain curve plot measures how well the model score sorts the data compared to the true outcome value [41]. When the predictions sort in exactly the same order, the relative Gini coefficient is 1. When the model sorts poorly, the relative Gini coefficient is close to zero, or even negative. The relative Gini scores were high for both our training set and testing set. (0.9840 and 0.9829, respectively, Fig 2B).

## Results

Of the 64 variables collected, 22 were retained for analysis after minimizing correlation (Appendix A2-A5 in S1 Appendix). Multiple imputation was used to correct for missingness (less than 2%) in two of the retained variables, neither of which appeared in the final model. Fifteen variables were significant in bivariate models in the first step of purposeful selection, and were included in the initial multivariate model. Eight variables were significant in the initial multivariate model and were retained in the final model. Including variables that were non-significant in the bivariate models with these eight variables did not significantly change the performance of the model, as determined by the Likelihood Ratio Test. No potential

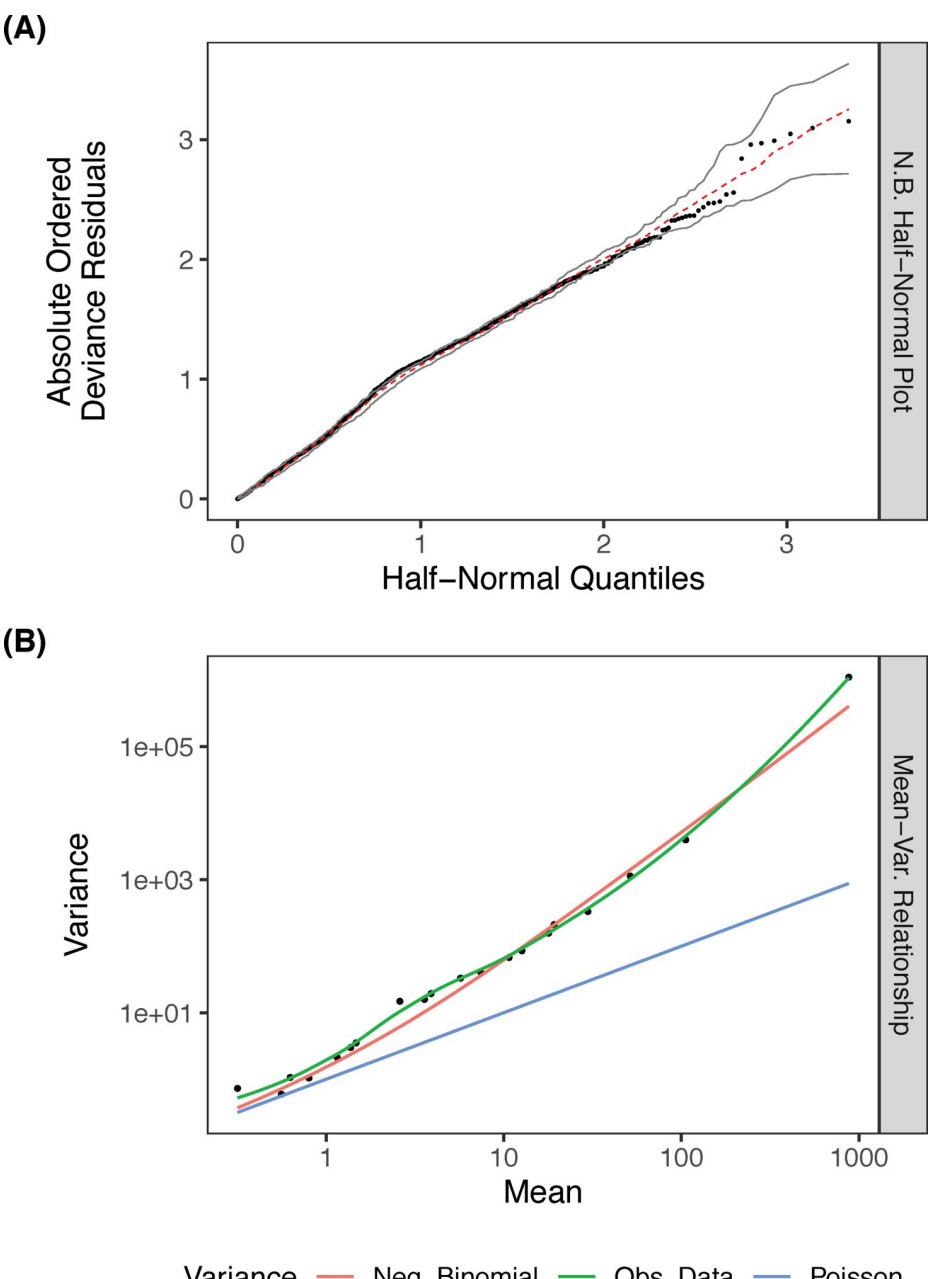

**Fig 1. Assessing model fit.** Plots showing (A) half-normal residuals and (B) mean-variance relationship of the observed county-level COVID-19 laCFRs.

confounders were identified among the correlation minimized variables that were previously discarded due to non-significance in the models.

The final model is shown in Table 1. The negative binomial model appears to be a good fit, capturing the mean-variance relationship observed in the data and displaying expected residuals (Fig 1A and 1B). The model was well-calibrated, with the training and testing model having comparable coverage and relative Gini score (Fig 2A and 2B). Since we used a negative binomial model with an offset, the exponentiated coefficients represent the change in laCFR observed for a one-unit increase in each continuous variable, assuming all other variables in the model are

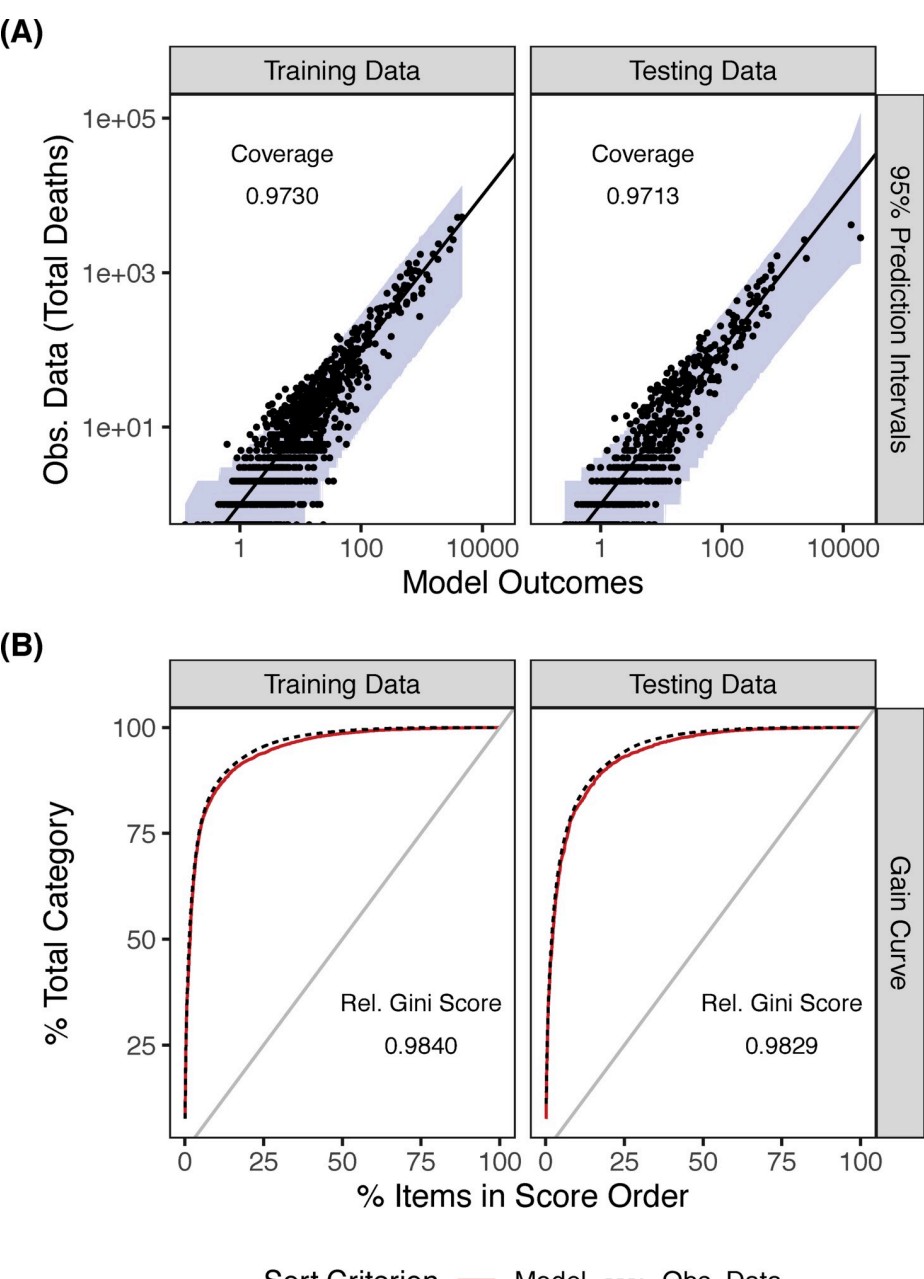

**Fig 2. Assessing model generality.** Plots showing (A) model outcomes found within the prediction intervals for training data and testing data for the county-level COVID-19 laCFRs and (B) gain curves for training data and testing data for the county-level COVID-19 laCFRs.

held constant. Four variables were inversely associated with laCFR: number of hospitals per 10,000 people (-32% laCFR per additional hospital per 10,000), banning religious gatherings during the initial state or county shutdown (-13% laCFR if religious gatherings were banned), percentage of housing units that were mobile homes (-0.79% laCFR per 1% increase in the proportion of mobile homes), and percentage of population without health insurance (-1.5% laCFR per 1% increase in percentage uninsured). Four variables were directly associated with laCFR: percentage over age 65 (+4.5% laCFR per 1% increase in population over age 65), percentage

**Table 1. Parameter estimates for the final multivariate model of laCFR.**

| Variable | Coefficient | 95% CI | p-value |
|---|---|---|---|
| Intercept | -4.5 | (-5.1, -3.9) | <0.001 |
| Percentage population aged 65+ | 0.044 | (0.030, 0.059) | <0.001 |
| Percentage population Black or African American | 0.0097 | (0.0063, 0.013) | <0.001 |
| Hospitals per 10,000 persons | -0.39 | (-0.59, -0.19) | <0.001 |
| Asthma prevalence | 0.091 | (0.039, 0.14) | <0.001 |
| Total number of hospitals | 0.031 | (0.0099, 0.054) | 0.0017 |
| Ban on religious gatherings indicator | -0.13 | (-0.24, -0.030) | 0.011 |
| Percentage housing stock that were mobile homes | -0.0079 | (-0.015, -0.0011) | 0.024 |
| Percentage population without health insurance | -0.015 | (-0.029, -0.00021) | 0.052 |

Black or African American (BAA) (+0.97% laCFR per 1% increase in BAA population), percentage with asthma (+9.5% laCFR per 1% increase in asthma prevalence), and number of hospitals (+3.2% laCFR per one additional hospital). Fig 3 demonstrates the relationship between each variable and the laCFR over a range of values. We have stratified these variables further for comparison, and results can be found in Appendix A6 in S1 Appendix.

## Discussion

In this ecological study of mortality due to SARS-CoV-2 infection during the first wave of COVID-19 in the US, we found that county-level laCFR was significantly associated with eight variables. Four variables–banning religious gathering, proportion of mobile homes, hospitals per 10,000 persons, and proportion of uninsured individuals in a county–were associated with decreased laCFR. Four variables–percentage of population over age 65, total number of

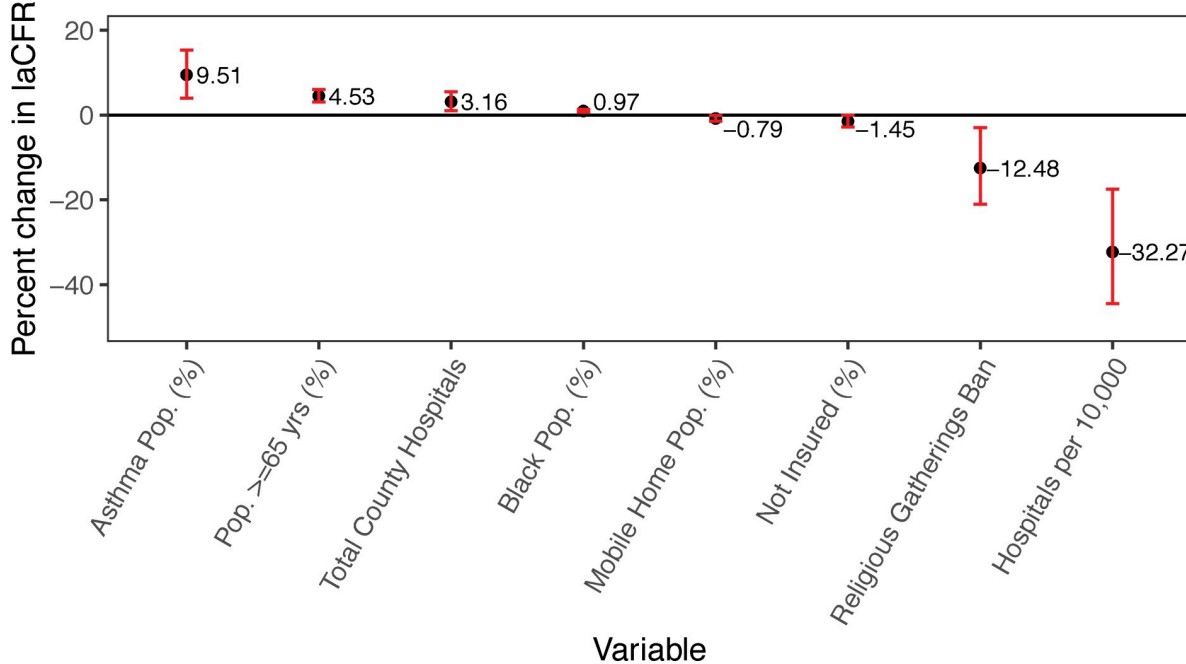

**Fig 3. Percentage change in COVID-19 laCFR given a 1 unit increase in the variable for each individual variable (shown in black dots) and 95% confidence interval (shown in red), using training data.**

hospitals per county, prevalence of asthma, and percentage of population BAA–were associated with increased laCFR. Each variable provided unique insights into factors that may be worth considering for county-level COVID-19 response efforts.

### Inverse association with case-fatality rate

Our model indicated a 13% reduction in the average laCFR for counties that banned religious gatherings compared to counties that did not. Gatherings often involve dense mixing of people in a confined space, sometimes over long periods of time [42], which drives COVID-19 transmission [43]. Interventions targeting increased physical distancing and limiting contact were introduced in some countries, including the closure of schools, places of worship, malls, and offices [42]. Our model suggests that specifically exempting religious gatherings from bans may increase the laCFR, consistent with the combination of findings that [1] religious gatherings across the globe were linked to COVID-19 superspreader events [44] and [2] older Americans (who are more likely to attend religious services than younger Americans [45]) are at increased risk of death due to COVID-19.

The percentage of the population living in mobile homes was also associated with a decrease in laCFR. A 1% increase in mobile home living was associated with a 0.79% decrease in laCFR. While a small difference at first glance, it becomes more meaningful when considering the large variation in mobile home living across counties. Between counties at the 25th percentile of percentage living in mobile homes (4%) and 75th percentile (18%), the difference in the percentage of mobile home living correlated with an 11% decrease in laCFR. This might represent a built-environment effect, given that mobile homes have separate plumbing and ventilation unlike apartments and other multi-family residences. Recent work suggests that fecal aerosol transmission of SARS-CoV2 can occur [46]. Ventilation patterns in apartment complexes represent additional opportunities for transmission [47]. The benefit of separate units such as mobile homes may be especially important to low-income workers who are both more at risk of death from COVID19 due to increased chance of having a co-morbid condition and more likely to live in multi-family housing with maintenance issues [48].

The number of hospitals per 10,000 was also inversely associated with laCFR. We found that for each additional hospital per 10,000 inhabitants, the laCFR decreased by 32%, despite the exclusion of other healthcare-related variables due to non-significance (e.g., ICU bed availability). Prior work demonstrated that the percentage of ICU and non-ICU beds occupied by COVID-19 patients directly correlated with COVID-19 deaths [49] and a county with more hospitals per 10,000 inhabitants may be able to cope with more COVID-19 cases before reaching the same percentage of hospitals beds occupied as a county with fewer hospitals per 10,000 inhabitants. Furthermore, because adding beds requires fewer resources than adding hospitals, the number of hospitals per 10,000 persons in a county might represent a greater ability to expand capacity. As a result, using hospitals per 10,000 may be a better indicator of healthcare capacity than the number of ICU beds early on in the pandemic. Because healthcare resources in the US correlate with community wealth [50], the rate of hospitals per 10,000 may also reflect increased community wealth and the protective effect of higher socio-economic status on health. More hospitals per 10,000 persons may also represent increased competition for patients, which is associated with decreased mortality from community-acquired pneumonia [51].

Unexpectedly, the percentage uninsured was inversely associated with laCFR. We found a 1.5% reduction in laCFR for every 1% increase in uninsured inhabitants. Prior studies found longer travel times to COVID-19 testing facilities were directly associated with percentage uninsured [52, 53]. Because uninsured persons may be unable to readily access testing, this

finding may relate to incomplete reporting, such that only individuals who survive long enough are tested for COVID-19, leading to a potential undercount of deaths attributable to SARS-CoV-2 infection.

### Direct association with case-fatality rate

In our model, a 1% increase in the population over 65 years old was associated with a 4.5% increase in average laCFR. This is consistent with recent epidemiological studies demonstrating an association between the severity of COVID-19 infection and age. According to provisional death data from the National Center of Health Statistics, people aged 65 and older have a 90- to 630-fold higher risk of mortality due to COVID-19 than 18-29-year olds [54].

Also, directly associated with laCFR was the total number of hospitals per county, with an observed increase of 3.2% in average laCFR per additional hospital. This variable was strongly correlated with total population (r = 0.92). Given that the number of hospitals per 10,000 was associated with decreased laCFR, this correlation suggests that total hospitals might be a proxy indicator for total population. Previous work assessed population density as a risk factor for increased laCFR, but not total population [43]. Since our analysis focused on the first wave of COVID-19, this variable could reflect overwhelmed healthcare systems in highly populated counties where most of the COVID-19 cases initially occurred [55].

Asthma prevalence was also directly associated with laCFR. A 1% increase in asthma prevalence was associated with a 9.5% increase in laCFR. Evidence regarding asthma as a risk factor in COVID-19 is mixed. Although the US CDC has determined that patients with moderate to severe asthma belong to a high-risk group [56], the Chinese CDC indicated that asthma was not a risk factor for severe COVID-19 [57]. One study showed that COVID-19 patients with asthma were of older age and had an increased prevalence of multiple comorbidities compared to those without asthma [58], but that the presence of asthma alone was not a risk factor for increased mortality [58]. Thus, despite our findings, it is unclear whether asthma has a direct impact on COVID-19 disease or if other factors may be associated with both asthma and COVID-19. One such potential confounder is exposure to air pollution, as air pollution is associated with both asthma and risk of death from COVID-19 [59].

Finally, laCFR was directly associated with the percentage of the population identifying as BAA in a county. Our model showed that a 1% increase in BAA was associated with a 0.97% increase in the laCFR. This likely reflects the effects of structural racism in the US, where BAAs have fewer economic and educational opportunities than White Americans and as a result are exposed to increased risk of morbidity and mortality from COVID-19 [60]. Dalsania et al. also found that the social determinants of health contributed to an unequal impact of the COVID-19 pandemic for BAA at the county level [61]. A study by Golestaneh showed that US counties with BAA as the majority had three times the rate of infection and almost six times the rate of death as majority White counties [62]. Factors underlying this trend include years of structural racism resulting in a lack of financial resources, increased reliance on public transportation, housing instability, and dependence on low-paying retail jobs [63]. Our approach considered several other variables that might explain the effect but were either non-significant (e.g., household crowding, percentage of households without a vehicle, and county land area) or were correlated with percentage BAA (e.g. percentage single parent households and percentage living in poverty), further emphasizing the role of systemic racism in COVID-19 laCFR.

### Excluded predictor variables

In reducing multicollinearity and using purposeful selection, several variables were surprisingly excluded. One of these excluded variables was population density, although higher

population density had been hypothesized to increase contact rate and non-adherence with physical distancing [43]. Diabetes and cardiovascular disease were excluded, despite multiple studies reporting these conditions as risk factors for COVID-19 mortality [64, 65]. While these factors are important at an individual level to assess the mortality risk, our model suggests that other variables may be more informative at the county-level, underscoring the value of ecological studies.

## Study strengths and limitations

This study had several strengths besides the benefits of an ecological design when considering population interventions. First, the data were nationally representative, including over 50% of all US counties. Our model captured the variability in the data and accounted for the observed data distribution. The model also captured almost all outcomes within the prediction interval for both training and testing data sets, with similar accuracy between them, which indicates that our model is generalizable within the US. Additionally, our model based laCFR calculations on the distribution of times from hospitalization to death from US data [35], which differed from earlier Chinese data [57]. Using US-based distribution of times likely improved our laCFR estimation for this study. The final model included several variables previously attributed to higher laCFR (such as older age) [54] and included a variable unique to the pandemic shutdown, i.e., banning religious gatherings, giving more nuanced insights into heterogeneous COVID-19 mortality rates across counties.

Despite these strengths, our study had several limitations. First, under-reporting of cases might affect the accuracy of CFR calculation [66]. The reported cases and deaths we used likely underestimated the true COVID-19 parameters. This underestimation was more among the asymptomatic and mild cases due to limited testing capacity and changes in testing practice; hence, the laCFR might have appeared inflated. Second, the type and timing of the tests used may have impacted the measured laCFR. Samples collected early during the infection can yield higher false negatives with RT-PCR tests [67]. False negatives in critically-ill patients who later die could decrease the measured laCFR unless probable COVID-19 deaths are reported, while false negatives in mild cases who are not retested later could increase the measured laCFR as survivable cases go undetected. These are challenges for any CFR study and highlight the ongoing need for improved COVID-19 testing. Third, COVID-19 reporting practices vary widely by state. For example, Florida was found to report fewer COVID-19 deaths in the official tally than the Medical Examiners Commission [68]. In addition to deliberate underreporting of deaths, states also vary in reporting of probable cases and deaths [69]. Without national standards in the COVID-19 response, comparing case counts and deaths across state line–let alone county–is deterred by lack of clarity about how these data differ [69].

Beyond these, our study was also limited by the fact that relevant data were frequently unavailable, including data on non-pharmaceutical interventions (NPI) and comorbidities. To limit missingness in the NPI data, we used state-level data when available given that counties also enforce state-level orders. However, there may be heterogeneity between county- and state-level information making this a less effective approach. Other variables of interest were not available at the state- or county-level, including information on contact tracing efforts and community compliance with public health mandates. Funding to collect public health information on more variables at a granular level would improve the information available to guide decision-making during emergencies. Another limitation was the highly correlated nature of the 64 variables considered for inclusion. Multicollinearity greatly affects the interpretability of coefficients and is rarely accounted for in epidemiologic studies [70]. Highly correlated variables in a model are unstable and can bias standard errors, leading to unreliable p-values and

unrealistic interpretations [70]. Because we ensured our model interpretability by excluding highly correlated variables, not all of our collected 64 variables were screened for inclusion in the final model.

Finally, our study period ended in mid-June. Recent work has divided the COVID-19 pandemic in the USA into three waves, with the first wave running from late March 2020 until mid-June 2020 [71]. The exact day of June 12, 2020 was chosen because [1] enough cases had occurred in the US to obtain reliable estimates of laCFR by county and [2] it preceded CDC reopening guidance and a shift in reporting to the HHS Protect system, which is less readily available to the public than the prior CDC reporting system [72]. The decision by the government to switch to the HHS Protect system hinders the ability of academic scientists to aid in the response to the on-going pandemic [72]. Making these data more readily available to the public would permit inclusion of additional data for future research.

## Conclusion

This study highlights several variables that were associated with county-level laCFR during the first wave of COVID-19 in the US. Though further research is needed to examine the effects of additional NPIs, our work provides insights that may aid in targeting response and vaccination efforts for improved outcomes in subsequent waves.

## Supporting information

**S1 Appendix.**
(PDF)

## Acknowledgments

This project is part of the COVID-19 Dispersed Volunteer Research Network (COVID-19-DVRN) led by M. Majumder and A. Desai. The authors thank Marie Charpignon, Catherine Pollack, and Emily Ricotta for their thoughtful feedback and review of the manuscript.

## Author Contributions

**Conceptualization:** Jess A. Millar, Hanh Dung N. Dao, Marianne E. Stefopulos, Camila G. Estevam, Katharine Fagan-Garcia, Diana H. Taft, Christopher Park, Angel N. Desai, Maimuna S. Majumder.

**Data curation:** Jess A. Millar, Hanh Dung N. Dao, Marianne E. Stefopulos, Camila G. Estevam, Katharine Fagan-Garcia, Diana H. Taft, Christopher Park, Amaal Alruwaily.

**Formal analysis:** Jess A. Millar.

**Investigation:** Jess A. Millar, Hanh Dung N. Dao, Marianne E. Stefopulos, Camila G. Estevam, Katharine Fagan-Garcia, Diana H. Taft, Christopher Park, Amaal Alruwaily, Angel N. Desai, Maimuna S. Majumder.

**Methodology:** Jess A. Millar, Hanh Dung N. Dao.

**Project administration:** Diana H. Taft.

**Software:** Jess A. Millar, Hanh Dung N. Dao.

**Supervision:** Angel N. Desai, Maimuna S. Majumder.

**Validation:** Jess A. Millar.

**Visualization:** Jess A. Millar.

**Writing – original draft:** Jess A. Millar, Hanh Dung N. Dao, Marianne E. Stefopulos, Camila G. Estevam, Katharine Fagan-Garcia, Diana H. Taft, Christopher Park, Amaal Alruwaily.

**Writing – review & editing:** Jess A. Millar, Hanh Dung N. Dao, Marianne E. Stefopulos, Camila G. Estevam, Katharine Fagan-Garcia, Diana H. Taft, Christopher Park, Amaal Alruwaily, Angel N. Desai, Maimuna S. Majumder.

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
