## [Decision Letter · Decision Letter 0]

7 May 2021

PONE-D-21-07386

Risk factors for increased COVID-19 case-fatality in the United States: A county-level analysis during the first wave

PLOS ONE

Dear Dr. Millar,

Thank you for submitting your manuscript to PLOS ONE. After careful consideration, we feel that it has merit but does not fully meet PLOS ONE’s publication criteria as it currently stands. Therefore, we invite you to submit a revised version of the manuscript that addresses the points raised during the review process.

We look forward to receiving your revised manuscript.

Kind regards,

Wen-Jun Tu

Academic Editor

PLOS ONE

Journal Requirements:

2. In your Methods section, please provide sufficient information to make the study reproducible. Please ensure that you have reported the equations representing the model; how the model was calibrated; and what parameters were applied.

 [The funders had no role in study design, data collection and analysis, decision to publish, or preparation of the manuscript.].

4. Thank you for submitting the above manuscript to PLOS ONE. During our internal evaluation of the manuscript, we found significant text overlap between your submission and the following previously published works, some of which you are an author.

https://www.sciencedirect.com/science/article/pii/S0306453021001190?via%3Dihub

Please revise the manuscript to rephrase the duplicated text, cite your sources, and provide details as to how the current manuscript advances on previous work. Please note that further consideration is dependent on the submission of a manuscript that addresses these concerns about the overlap in text with published work.

Reviewers' comments:

Reviewer's Responses to Questions

**Comments to the Author**

1. Is the manuscript technically sound, and do the data support the conclusions?

Reviewer #1: Yes

Reviewer #2: Partly

Reviewer #3: Yes

2. Has the statistical analysis been performed appropriately and rigorously? 

Reviewer #1: Yes

Reviewer #2: I Don't Know

Reviewer #3: Yes

3. Have the authors made all data underlying the findings in their manuscript fully available?

Reviewer #1: Yes

Reviewer #2: No

Reviewer #3: Yes

4. Is the manuscript presented in an intelligible fashion and written in standard English?

Reviewer #1: Yes

Reviewer #2: Yes

Reviewer #3: Yes

5. Review Comments to the Author

Reviewer #1: The ongoing COVID-19 pandemic is causing significant morbidity and mortality across the US. The distribution of infected cases and fatalities in the US has been heterogeneous across counties, and identification of sub-populations at risk of increased morbidity and mortality remains crucial to effective response efforts by federal, state, and local governments. In this ecological study, we used a lag-adjusted COVID-19 case-fatality rate (CFR) to conduct a county-level mortality risk factor analysis of demographic, socioeconomic, and health-related variables in the US during the first wave of the COVID-19 pandemic (March 28, 2020 to June 12, 2020). We identified county level variables associated with the COVID-19 CFR using publicly available datasets and a negative binomial generalized linear model. Variables associated with decreased CFR included a greater number of hospitals per 10,000 people, banning religious gatherings, a higher percentage of people living in mobile homes, and a higher percentage of uninsured people. Variables associated with increased CFR included a higher percentage of the population over age 65, a higher percentage of Black or African Americans, a higher asthma prevalence, and a greater number of hospitals in a county. By identifying factors that are associated with COVID-19 CFR in US counties, our work provides insights that may help officials target public health interventions and healthcare resources to locations that are at increased risk of COVID-19 fatalities in subsequent waves

Comments：

1. “By June 12th, 2020, SARS-CoV-2 had caused over 2 million Coronavirus Disease 2019 (COVID-19) cases and 114,753 deaths in the United States (US) (2,3)” please update this data

2. The Research hypotheses and objective need to be clear in the introduction section

3. Study Population, Research flow chart needs to be provided. How many patients included and excluded, why?

4. How potential risk factors were identified? Why those factors were included.

5. Results, “retained variables, neither of which appeared in the final model (21)” reference should not be included in the results section.

6. a county-level mortality could be compared between March 28, 2020 to June 12, 2020 vs. March 28, 2019 to June 12, 2019?

7. What is the clinical utility of the authors findings? The clinical perspective should be confirmed. The findings could be used to reduce the COVID-19 case-fatality in the United States?

8. In Discussion section, the authors mainly summarized again their observations and how they were similar or different from prior studies. This reviewer would expect to see some points regarding how to translate these observations to help address this public health concern.

Minor points:

Added the following references in the revision text “Tu, WJ., Cao, J., Yu, L. et al. Clinicolaboratory study of 25 fatal cases of COVID-19 in Wuhan. Intensive Care Med 46, 1117–1120 (2020). https://doi.org/10.1007/s00134-020-06023-4; Cao, J., Tu, W. J., Cheng, W., Yu, L., Liu, Y. K., Hu, X., & Liu, Q. (2020). Clinical features and short-term outcomes of 102 patients with coronavirus disease 2019 in Wuhan, China. Clinical Infectious Diseases, 71(15), 748-755.”

Reviewer #2: This manuscript does not bring new useful information to readers, so it may not get more attention from readers. The manuscript is less innovative. It is recommended to add relevant analysis indicators and data to improve the quality.

Reviewer #3: This paper has clear ideas and reliable statistical data.Sufficient research subjects are included.According to the current situation of COVID-19, it is very important to carry out related research.Further reports on the second wave of the pandemic can be made.

But there are some problems as follows.

1.The first wave of the COVID-19 pandemic defined from March 28, 2020 to June 12, 2020.But how is the first wave of a pandemic defined.The reasons given in the article do not seem particularly convincing.

2.Some variables can be further stratified for comparison.The more detailed analysis of these variables might lead to more precise conclusions.

3.Some of the images should be further simplified and beautified.It can better improve the readability of the paper.

6. PLOS authors have the option to publish the peer review history of their article (what does this mean?). If published, this will include your full peer review and any attached files.

Reviewer #1: No

Reviewer #2: No

Reviewer #3: No

---

## [Author Response · Author response to Decision Letter 0]

21 Aug 2021

Dear Editor,

Thank you for the opportunity to revise and resubmit our manuscript. Below is a point-by-point response to the reviewer comments. Addressing these comments has significantly strengthened our manuscript, and we look forward to next steps in the review process.

Reviewer #1, comment 1: “By June 12th, 2020, SARS-CoV-2 had caused over 2 million Coronavirus Disease 2019 (COVID-19) cases and 114,753 deaths in the United States (US) (2,3)” please update this data

Response: Thank you for this comment. We opted to use this date as our work only considered cases during the first wave of the pandemic in the U.S. (i.e., through June 12, 2020). We have clarified this point and added additional information on the burden of COVID-19 in lines 49–50. 

Reviewer #1, comment 2: The Research hypotheses and objective need to be clear in the introduction section

Response: Thank you for letting us know that this was unclear. We have rephrased the last paragraph of the introduction (lines 74, 76–78) to highlight that our study objective was to use a lag-adjusted case fatality rate (laCFR) to determine which population-level variables were most associated with differences in laCFR across U.S. counties. 

Reviewer #1, comment 3: Study Population, Research flow chart needs to be provided. How many patients included and excluded, why?

Response: Thank you for this helpful suggestion. As this study was ecological in design, we did not enroll individual patients. Instead, our unit of analysis was U.S. counties. We have therefore added a flowchart in our supplemental materials indicating our county inclusion process (Appendix 1). Of 3,413 total U.S. counties, 3,004 counties had experienced at least one COVID-19 case by the end of our study window, June, 12, 2020. Only 1,779 counties reached a stable laCFR by the end of our study window. These counties were split into a training set (1,186 counties) and a testing set (593 counties).

Reviewer #1, comment 4: How potential risk factors were identified? Why those factors were included.

Response: Thank you for raising this need for clarification. Our process of variable inclusion had 2 steps: (1) an identification of the initial cohort of variables and (2) the variable selection for the regression model. For the first step, we identified potential county-level risk factors through a comprehensive literature review, which we further supplemented with variables relevant to other respiratory epidemics. In our original submission, we provided a detailed description and justification for the inclusion of the initial cohort of variables in Appendices 2–4. For the second step, as described in the main text of our original submission, we removed highly correlated variables to reduce multicollinearity and used purposeful selection to identify variables to be kept in the regression model. Since the first step was not clearly described in the main text, we have edited and added more details on the inclusion of these variables in the Methods section (lines 101–106).

Reviewer #1, comment 5: Results, “retained variables, neither of which appeared in the final model (21)” reference should not be included in the results section.

Response: Thank you for catching this oversight. We have moved the citation to the Methods section (line 127).

Reviewer #1, comment 6: a county-level mortality could be compared between March 28, 2020 to June 12, 2020 vs. March 28, 2019 to June 12, 2019?

Response: Thank you to the reviewer for this interesting suggestion on examining excess mortality at the county level. Unfortunately, our work relies on publicly available data sources and complete data on all-cause mortality for 2020 is not yet available from the National Vital Statistics System (i.e., the gold standard source for mortality data in the U.S.) at time of writing this response letter. As such, an excess mortality analysis comparing 2019 and 2020 is not possible in the present work. We will keep this suggestion in mind for future work, as sufficiently complete mortality data become publicly available.

Reviewer #1, comment 7: What is the clinical utility of the authors findings? The clinical perspective should be confirmed. The findings could be used to reduce the COVID-19 case-fatality in the United States?

Response: We would like to thank the reviewer for reminding us of the importance of a clinical perspective during the ongoing pandemic. Notably, this work is not designed to provide insight to clinicians seeking to care for individual patients; instead, this work uses an ecological study design and therefore considers the laCFR in a county as whole, without linking specific cases and deaths to specific individuals. Such a design is an excellent approach for public (i.e., population-level) health problems, however––and as a result, our work aims to provide decision-making support for public health officials who seek to understand which counties are most at risk of high rates of COVID-19 deaths. Investing in public health in such a way helps individual patients by informing targeted resource deployment such as vaccines, non-pharmaceutical interventions, and extra healthcare workers and supplies to the counties that need them most.

Reviewer #1, comment 8: In Discussion section, the authors mainly summarized again their observations and how they were similar or different from prior studies. This reviewer would expect to see some points regarding how to translate these observations to help address this public health concern.

Response: Thank you for this suggestion. As this paper is not focused on the effectiveness of specific interventions, we sought to explain reasons why we observed the associations we did without going beyond the bounds of what our work reasonably allows us to infer. In the Discussion section, we sought to contextualize our results but also felt it was important to avoid making specific policy recommendations that could not be supported directly from the present work. Therefore, to contextualize our results, we presented possible reasons for each of the observed relationships, supported by additional literature, especially when our results differed from the results seen in individual-level studies. Our finding that more hospitals per 10,000 persons were associated with lower laCFR is one example of such a result. In the Discussion section, we connect this finding to several possible reasons why this might be so: increased availability of hospital beds, increased ability to expand capacity rapidly, increased community wealth, and increased competition between hospitals leading to better outcomes for patients. Because our study was not designed to test which of these possible reasons caused the observed association, it would be an overstatement to make policy recommendations based on this observation. We faced a similar dilemma for each of the variables included in our model, so we opted to present well-researched potential reasons for their association with laCFR without providing specific recommendations. Instead, we suggest that these are factors worth considering but leave policy decisions to stakeholders to avoid overstating our findings. 

Reviewer #1, minor comment: Added the following references in the revision text “Tu, WJ., Cao, J., Yu, L. et al. Clinicolaboratory study of 25 fatal cases of COVID-19 in Wuhan. Intensive Care Med 46, 1117–1120 (2020). https://doi.org/10.1007/s00134-020-06023-4; Cao, J., Tu, W. J., Cheng, W., Yu, L., Liu, Y. K., Hu, X., & Liu, Q. (2020). Clinical features and short-term outcomes of 102 patients with coronavirus disease 2019 in Wuhan, China. Clinical Infectious Diseases, 71(15), 748-755.”

Response: Thank you for this suggestion. We have added these citations to the introduction (lines 54–57).

Reviewer #2, only comment: This manuscript does not bring new useful information to readers, so it may not get more attention from readers. The manuscript is less innovative. It is recommended to add relevant analysis indicators and data to improve the quality.

Response: Thank you to the reviewer for taking the time to read our work. We respectfully disagree with this assessment, and further feel that our work is scientifically sound and therefore eligible for publication in PLoS One. We believe that the careful consideration of factors associated with COVID-19 laCFR at the population level is innovative and of interest to public health officials, particularly because the factors previously associated with risk of COVID-19 at the individual level did not always match the factors identified in our model at the population level. Given that both clinical (i.e., individual-level) and public health (i.e., population-level) decision-making is essential to mitigating a pandemic, thus necessitating both types of analyses. We would gladly incorporate additional information to strengthen our paper, as we have sought to do for the other reviewer comments, but we were unable to determine what additional indicators and data were viewed as necessary to improve this work from the feedback provided here.

Reviewer #3, comment 1: The first wave of the COVID-19 pandemic defined from March 28, 2020 to June 12, 2020.But how is the first wave of a pandemic defined. The reasons given in the article do not seem particularly convincing.

Response: Thank you for letting us know that we did not provide adequate support for our selection of dates for the first wave. To address this concern, we have added a citation (lines 358–360) to a paper that discusses the three waves of the COVID-19 pandemic, a spring 2020 wave, a summer 2020 wave, and a fall/winter 2020/2021 wave. The paper does not give exact start and end dates for the waves, but Figure 1a plots cases over time and it appears that the first “spring” wave begins towards the end of March 2020 and ends at some point in mid-June 2020. We hope that this additional citation, combined with the other reasons previously presented in our manuscript, adequately defends our choice of dates.

Reviewer #3, comment 2: Some variables can be further stratified for comparison. The more detailed analysis of these variables might lead to more precise conclusions.

Response: Thank you for this suggestion for stratification. We have added the stratification results in the Appendix A9 and have also mentioned them in the Results section (lines 172–173).

Reviewer #3, comment 3: Some of the images should be further simplified and beautified. It can better improve the readability of the paper.

Response: Thank you for pointing this out. After careful consideration of our figures, we determined that Figure 1 was the figure that needed simplification and beautification. We have therefore split what was Figure 1 into two separate figures, to allow space for clearer legends. We are happy to make additional changes but felt that the remaining figures were already sufficiently simple and clear.

Reviewer's Responses to Question 3: Have the authors made all data underlying the findings in their manuscript fully available? 

Reviewer #1: Yes

Reviewer #2: No

Reviewer #3: Yes

Response: For the purposes of reproducibility, all data, code, and model calibration specifications (i.e., which seeds were used, etc.) for our paper were deposited in our GitHub repo (https://github.com/jmillar201/COVID19_CFR) at the time of our original submission.

Thank you for your time and consideration,

The Authors

---

## [Decision Letter · Decision Letter 1]

21 Sep 2021

PONE-D-21-07386R1Risk factors for increased COVID-19 case-fatality in the United States: A county-level analysis during the first wavePLOS ONE

Dear Dr. Millar,

Thank you for submitting your manuscript to PLOS ONE. After careful consideration, we feel that it has merit but does not fully meet PLOS ONE’s publication criteria as it currently stands. Therefore, we invite you to submit a revised version of the manuscript that addresses the points raised during the review process.

We look forward to receiving your revised manuscript.

Kind regards,

Wen-Jun Tu

Academic Editor

PLOS ONE

Journal Requirements:

Additional Editor Comments (if provided):

Please be clear about the innovative nature of this research.

Reviewers' comments:

Reviewer's Responses to Questions

**Comments to the Author**

1. If the authors have adequately addressed your comments raised in a previous round of review and you feel that this manuscript is now acceptable for publication, you may indicate that here to bypass the “Comments to the Author” section, enter your conflict of interest statement in the “Confidential to Editor” section, and submit your "Accept" recommendation.

Reviewer #1: All comments have been addressed

Reviewer #2: All comments have been addressed

2. Is the manuscript technically sound, and do the data support the conclusions?

Reviewer #1: Yes

Reviewer #2: Partly

3. Has the statistical analysis been performed appropriately and rigorously? 

Reviewer #1: Yes

Reviewer #2: Yes

4. Have the authors made all data underlying the findings in their manuscript fully available?

Reviewer #1: Yes

Reviewer #2: Yes

5. Is the manuscript presented in an intelligible fashion and written in standard English?

Reviewer #1: Yes

Reviewer #2: Yes

6. Review Comments to the Author

Reviewer #1: Congratulations to the authors for their excellent work and I am happy with their attention to the points raised.

Reviewer #2: I have no more comments about this manuscript "Risk factors for increased COVID-19 case-fatality in the United States: A county-level analysis during the first wave" submitted to Plos One.

7. PLOS authors have the option to publish the peer review history of their article (what does this mean?). If published, this will include your full peer review and any attached files.

Reviewer #1: No

Reviewer #2: No

---

## [Author Response · Author response to Decision Letter 1]

22 Sep 2021

Dear Editor,

Thank you for the opportunity to revise and resubmit our manuscript. Below is a point-by-point response to the reviewer comments. Addressing these comments has significantly strengthened our manuscript, and we look forward to next steps in the review process.

Editor, comment 1: Please review your reference list to ensure that it is complete and correct.

Response: We have checked all of the citations and replaced two preprints (details below). We have also provided archived links for several non-journal citations (such as research from the CDC) to make sure these links are stable over time. Here is the reasoning for citations that are not journal articles or book chapters:

Citation 2:

Citing where COVID19 cases and death data came from

Citation 4:

Citing where estimation of total mortality due to COVID-19 is located

Citation 13:

Citing our code that we used for this project on GitHub

Citation 17:

Citing where county level race data was acquired

Citation 18:

Citing where county level ICU beds and percent insured data was acquired

Citation 21-23:

Citation of work being done with the CDC on social vulnerability

Citation 25:

Citing where county level ICU bed data was acquired

Citation 26:

Citing where county level heart disease data was acquired

Citation 27:

Citing where county level diabetes data was acquired

Citation 28:

Citing where county level smoking and asthma data was acquired

Citation 29:

Citing where county level cancer data was acquired

Citation 30:

Citing where county level COVID19 non pharmaceutical intervention data was acquired

Citation 31:

Citing CDC reopening guidelines

Citation 32:

Citing where New York City county level COVID19 cases and death data came from

Citation 41:

This was a preprint citation that covered gain curve plots and their uses pretty well. Since this is a preprint, we went ahead and replaced this citation with a book chapter instead, covering the same topic

Citation 48:

Citing a policy brief released by the Community and Labor Center on the effect of COVID-19 on workers

Citation 54:

Citing National Center of Health Statistics on risk for age related mortality due to COVID-19

Citation 56:

Citing CDC on risk for asthma related mortality due to COVID-19

Citation 60:

Citing National Association of Counties (organization that represents county governments) on effects of structural racism in the US

Citation 68:

Citing news article on how Florida was found to report fewer COVID-19 deaths in the official tally than the Medical Examiners Commission

Citation 71:

This was a preprint citation that covered the different waves of the pandemic. Since this is a preprint, we have replaced this with a different article that is published

Citation 72:

Citation covering the shift in reporting to the HHS Protect system, as of June 12th

Editor, comment 2: Please be clear about the innovative nature of this research.

Response: Thank you for letting us know that this was unclear. We have expanded upon the innovative nature of the paper in the introduction. This work expands upon prior work by considering possible risk factors of increased mortality from multiple categories at once (e.g., non-pharmaceutical interventions such as shelter-in-place orders, prevalence of pre-existing conditions such as cardiovascular disease, and socio-economic circumstances such as hospital accessibility) in a single model. This is also the first paper to focus on this range of risk factors during the first wave of the pandemic, so that results from this can be used for targeted intervention at the county level at the beginning of a pandemic.

Reviewer #2, comment 1: Is the manuscript technically sound, and do the data support the conclusions?: Partly

Response: Reviewer 2 felt that either our manuscript was not technically sound or that the data do not support the conclusions we drew. However, reviewer 2 provided no comment on what the remaining issues were. We do feel our work is sound and our conclusions were appropriate, so are uncertain what to do to resolve this issue.

Thank you for your time and consideration,

The Authors

---

## [Editor Report · Decision Letter 2]

24 Sep 2021

Risk factors for increased COVID-19 case-fatality in the United States: A county-level analysis during the first wave

PONE-D-21-07386R2

Dear Dr. Millar,

We’re pleased to inform you that your manuscript has been judged scientifically suitable for publication and will be formally accepted for publication once it meets all outstanding technical requirements.

Kind regards,

Wen-Jun Tu

Academic Editor

PLOS ONE
---

## [Editor Report · Acceptance letter]

28 Sep 2021

PONE-D-21-07386R2 

Risk factors for increased COVID-19 case-fatality in the United States: A county-level analysis during the first wave 

Dear Dr. Millar:

I'm pleased to inform you that your manuscript has been deemed suitable for publication in PLOS ONE. Congratulations! Your manuscript is now with our production department. 

Kind regards, 

on behalf of

Dr. Wen-Jun Tu 

Academic Editor

PLOS ONE